# Genome-Wide Analysis of Aux/IAA Gene Family in *Artemisia argyi*: Identification, Phylogenetic Analysis, and Determination of Response to Various Phytohormones

**DOI:** 10.3390/plants13050564

**Published:** 2024-02-20

**Authors:** Conglong Lian, Jinxu Lan, Rui Ma, Jingjing Li, Fei Zhang, Bao Zhang, Xiuyu Liu, Suiqing Chen

**Affiliations:** 1School of Pharmacy, Henan University of Chinese Medicine, 156 East Jin-shui Rd., Zhengzhou 450046, China; liancl00@163.com (C.L.); baozhang923@163.com (B.Z.); liuxiuyuzy@163.com (X.L.); 2Henan Key Laboratory of Chinese Medicine Resources and Chemistry, 156 East Jin-shui Rd., Zhengzhou 450046, China

**Keywords:** *Artemisia argyi*, Aux/IAA family, expression pattern, function prediction

## Abstract

*Artemisia argyi* is a traditional herbal medicine plant, and its folium artemisia argyi is widely in demand due to moxibustion applications globally. The Auxin/indole-3-acetic acid (Aux/IAA, or IAA) gene family has critical roles in the primary auxin-response process, with extensive involvement in plant development and stresses, controlling various essential traits of plants. However, the systematic investigation of the Aux/IAA gene family in *A. argyi* remains limited. In this study, a total of 61 Aux/IAA genes were comprehensively identified and characterized. Gene structural analysis indicated that 46 Aux/IAA proteins contain the four typical domains, and 15 Aux/IAA proteins belong to non-canonical IAA proteins. Collinear prediction and phylogenetic relationship analyses suggested that Aux/IAA proteins were grouped into 13 distinct categories, and most Aux/IAA genes might experience gene loss during the tandem duplication process. Promoter *cis*-element investigation indicated that Aux/IAA promoters contain a variety of plant hormone response and stress response *cis*-elements. Protein interaction prediction analysis demonstrated that AaIAA26/29/7/34 proteins are possibly core members of the Aux/IAA family interaction. Expression analysis in roots and leaves via RNA-seq data indicated that the expression of some AaIAAs exhibited tissue-specific expression patterns, and some AaIAAs were involved in the regulation of salt and saline-alkali stresses. In addition, RT-qPCR results indicated that AaIAA genes have differential responses to auxin, with complex response patterns in response to other hormones, indicating that Aux/IAA may play a role in connecting auxin and other hormone signaling pathways. Overall, these findings shed more light on AaIAA genes and offer critical foundational knowledge toward the elucidation of their function during plant growth, stress response, and hormone networking of Aux/IAA family genes in *A. argyi*.

## 1. Introduction

*Artemisia argyi* Lev. et Vant., a perennial herb with a strong aroma belonging to the genus *Artemisia* and the family Asteraceae, is widely distributed in Asian countries, such as China, Japan, and Korea (https://www.plantplus.cn/info/Artemisia%20argyi?t=foc; accessed on 5 December 2023). In China, *A. argyi* is widely distributed except in extremely arid or cold regions, and the main growth areas include Nanyang in Henan Province, Qichun in Hubei Province, and Anguo in Hebei Province. *A. argyi* is known as a medicinal plant. The dried leaf of *A. argyi* is a well-known traditional Chinese medicine (TCM) known as “folium artemisiae argyi” or “Chinese mugwort”, which is commonly used in moxibustion for the treatment of hemostasis, epistaxis, hemorrhage, inflammation, abdominal cold pain, uterine cold infertility, itches, and menstruation-related ailments throughout Asia [1]. Modern pharmacological research uncovered that folium artemisiae argyi can be applied as an antibacterial or antiviral treatment because of its rich volatile oils [2,3]. In addition, *A. argyi* can be used in food, cosmetics, and daily necessities. This versatility of *A. argyi* leads to a massive market demand. Although the current production of *A. argyi* is considerable, and the output of fresh folium artemisiae argyi in China reached 192,000 tons in 2020, its production still cannot meet the market demand. Therefore, it is critical to study the growth and development of *A. argyi*.

Plant hormones are critical for the regulation of plant growth and development in various stages, especially auxin. Many of these rely on the auxin signaling pathway, which determines the fate of plants from birth to death and plays essential roles during plant life cycles, including embryogenesis, cell division and elongation, stress response, and secondary metabolite biosynthesis [4]. Auxin/indole-3-acetic acid (Aux/IAA, or IAA) proteins are important components of the auxin signaling pathway and are established as principal targets of the auxin transport inhibitor response 1 (TIR1) as well as the similar proteins AUXIN RECEPTOR F-BOX/AFBs [5,6]. Aux/IAA genes encode short-lived nuclear proteins that are degraded via the 26S proteasome and can be ubiquitinated by binding with TIR1/AFB receptors, causing ARF homodimers to activate the transcription of auxin-responsive genes under high auxin levels [7,8]. Aux/IAA proteins contain four highly conserved domains: I, II, III, and IV. Domains I and II and domains III and IV are situated at the N-terminus and C-terminus, respectively, producing distinct roles of repression, degradation, or interaction [9,10]. Domain I contains an amphiphilic motif LxLxLx that is associated with TOPLESS (TPL)/TPL-related (TPR) corepressors and is required for the transcriptional inhibitory function of Aux/IAA proteins [11]. Domain II contains the GWPPV motif, which acts as the target sequence of Aux/IAA protein ubiquitination for degradation [12]. Domains III and IV combine with the ARF and contribute to the homo- and hetero-dimerization between Aux/IAA and ARF proteins [13]. The diversity of conserved structures and functions indicates functional differences among different Aux/IAA members.

In the past years, the biological functions of Aux/IAA family genes have been revealed. In particular, Aux/IAA plays a pivotal role in monocots and dicots by affecting the growth and development of different tissues. In *Arabidopsis thaliana*, *Aux/IAA1* and *Aux/IAA8* genes have many roles in the regulation of the development of diverse organs, such as seed germination, root development, rosette leaves, shoot tropisms, and inflorescence [14,15]. In rice, *OsIAA1*, *OsIAA4*, *OsIAA11*, and *OsIAA12* are involved in plant height and leaf angle, tiller angle and gravity tropism, lateral roots formation, and leaf angle, respectively [16,17,18]. Moreover, *Aux/IAA* genes are involved in adversity stress responses. In *A. thaliana*, *AtIAA5/AtIAA6/AtIAA19* mediate drought response by regulating glucosinolate levels [19]. In rice, *OsIAA9* and *OsIAA20* genes are significantly upregulated under high salt stress [20]. The overexpression of *OsIAA6* triggered noticeable drought resistance enhancements [21]. *OsIAA20* plays a key role in plant responses to drought and salt stresses through the ABA signal transduction pathway [22]. In addition, Aux/IAA genes also impact secondary metabolism, such as the biosynthesis of volatile compounds, including α-humulene, α-terpinene, β-caryophyllene, β-phellandrene, and γ-element [23].

Recently, with the release of reference genomes for different species, Aux/IAA gene families are being identified and analyzed in an increasing number of plant species through bioinformatics approaches, including model plants *A. thaliana* (29 members) [24], *Populus* (35 members) [25], and maize (40 members) [26], as well as medicinal plants such as *Salvia miltiorrhiza* (23 members) [27], *Dendrobium officinale* (14 members) [28], and *Boehmeria nivea* (16 members) [29]. However, the identification and functional information of Aux/IAA genes in medicinal plants are also limited and uncharacterized, especially in the *A. argyi* genome. The recent publication of the whole genome data of *A. argyi* enabled a massive progression in the comprehension of genome organization, which allowed for the identification of Aux/IAA genes [30]. Examining the role of Aux/IAA genes in *A. argyi* could provide an improved understanding of the functional attributes of this gene family in *A. argyi* and stimulate further research in related medicinal plant organisms. In this study, we endeavored to uncover genome-wide characterization and organization of Aux/IAA genes, predicting the related protein domains, architecture, GO annotation, promoter *cis*-elements, phylogenetic relationships, protein interaction, and expression patterns in different tissues, and response to various hormones. These results contributed to the clarification of Aux/IAA protein functions. They provided valuable information for further studies in *A. argyi* and other medicinal plants to uncover the functional roles of Aux/IAA genes in growth and progression.

## 2. Results

### 2.1. Identification of the Aux/IAA Genes in A. argyi

Based on biosequence analysis using profile hidden Markov models (HMMER) and Basic Local Alignment Search Tool for proteins (BLASTP) selection, and after determining the presence of the Aux/IAA (PF02309) domain using the Pfam, a total of 61 Aux/IAA members were identified in the genome data of *A. argyi* (Table 1) [30]. Identified genes were named from *AaIAA1* to *AaIAA61* based on the AaIAA location on the chromosomes (Figure 1). The analysis of physicochemical properties demonstrated that the predicted AaIAA proteins varied in size from 155 (AaIAA20, AaIAA52, and AaIAA61) to 1089 amino acids (AaIAA7) with molecular masses and theoretical isoelectric points ranging from 17.8 kDa (AaIAA5) to 121.9 kDa (AaIAA7) and from 4.96 (AaIAA47) to 9.18 (AaIAA8), respectively. The protein instability coefficient of Aux/IAA proteins ranged from 28.42 (AaIAA49) to 70.7 (AaIAA29), and most are unstable except 13 proteins, namely AaIAA6/11/16/17/22/23/33/48/49/50/54/55/56 proteins, as their instability indexes were below 40. The aliphatic index varies from 57.47 (AaIAA18, AaIAA25, and AaIAA57) to 88.81 (AaIAA54). The average hydropathicity varies from −0.863 (AaIAA49) to −0.248 (AaIAA45), indicating that these proteins are hydrophilic. In addition, all proteins are situated in the nucleus, suggesting that they are likely to be transcription factors and play essential roles in the nucleus. The sizable variability in fundamental physical and chemical characteristics indicates that they may possess functions in varied microenvironments and have diverse functions in plant growth and development (Table 1).

### 2.2. Chromosomal Distribution, Collinear Prediction, and Sequence Similarity Analysis of AaIAA Genes

According to the study by Hongyu Chen et al., *A. argyi* can be assembled into 17 pseudochromosomes on the chromosome scale [30]. A total of 57 *AaIAA* genes in *A. argyi* were distributed on 17 chromosomes (Chr), with *AaIAA58*, *AaIAA59*, *AaIAA60*, and *AaIAA61* distributed in the scaffolds of GWHBRAE00001633, GWHBRAE00001586, GWHBRAE00001446, and GWHBRAE00002123, respectively (Figure 1). The distribution of *AaIAAs* genes varies: there was just a single *AaIAA57* gene on Chr17; Chr10 had the largest number with eight total genes; Chr11 had seven genes; and the remaining 15 chromosomes were distributed with two to four genes. In addition, there are tandem duplication phenomena in the distribution of *AaIAA* genes, including *AaIAA5* and *AaIAA6* genes on Chr2, *AaIAA22* and *AaIAA23* genes on Chr8, and *AaIAA55* and *AaIAA56* genes on Chr16 (Figure 1).

Gene divergence and duplication events are critical causes of evolutionary momentum and can also significantly promote the expansion of gene families and the diversification of protein functionalities [31]. In this study, 57 IAA family gene collinear events were used to reveal the expansion of the *A. argyi* IAA family genes during evolution (Figure 2A). A total of 45 *AaIAA* genes have 106 collinear events. Among them, the majority of the *A. argyi AaIAA* genes had one (18 genes) or two (14 genes) collinear events, while 4, 2, 2, and 5 *AaIAA* genes had 3, 4, 5, and 6 collinear events, respectively (Figure 2A, Appendix A). These findings implied that homologous genes may exert similar functions, indicating functional redundancy or functional diversity in the Aux/IAA family in *A. argyi*. Furthermore, a visualization of the sequence similarity of *AaIAA* genes in *A. argyi* was analyzed by Circoletto [32]. As depicted in Figure 2B, 11 genes with red ribbons indicate that they have the highest similarity homologous genes, including *AaIAA21* to *AaIAA53*, *AaIAA7* to *AaIAA29*, *AaIAA1/27* to *AaIAA28*, and *AaIAA9*/*10*/*39* to *AaIAA60*. A total of 47 genes with orange ribbons indicate they also have 75% < orange ≤ 99% similarly homologous genes.

Additionally, we analyzed the collinear events of *AaIAA* genes within the *A. argyi* genome and between the *A. thaliana* genome (Figure 2C, Appendix A). Compared to *A. argyi* and *A. thaliana*, 35 *AaIAA* genes in *A. argyi* had orthologs with 21 members of *A. thaliana*. Most of the *A. argyi AaIAA* genes had one (13 genes) or two (13 genes) orthologs in the *A. thaliana* genome, seven *A. argyi AaIAA* genes each had three orthologs in *A. thaliana*, and another two *A. argyi AaIAA* genes had four orthologs in *A. thaliana*. In contrast, there were one to seven orthologs when comparing *A. thaliana* to *A. argyi* (Figure 2C, Appendix A). These findings implied that most *AaIAA* genes experienced gene loss in the tandem duplication process of the AaIAA gene family throughout *A. argyi*.

### 2.3. Gene Structure, Motif Composition, and Promoter Cis-Element Analysis of AaIAA Genes

Gene structure schematics of exon–intron organizations of all identified AaIAA genes were generated using GSDS, which can provide more insight into the evolution of the Aux/IAA family in *A. argyi* (Figure 3). The number of introns in all AaIAA genes was between 1 and 14. The coding sequences of most (67.2%) of the AaIAA genes were disrupted by three (nine AaIAAs) or four (32 AaIAAs) introns, and the intron locations and situations are highly conserved (Figure 3C). Subsequent assessments suggested that the dispersal and situations of introns coincided with the evolutionary alignment of all AaIAA genes (Figure 3A).

The Multiple Em for Motif Elicitation (MEME) web server, including 20 motifs, was used to analyze the conserved motif distributions of AaIAA proteins. The proteins of AaIAAs vary from 3 to 13 motifs, with most of the proteins having motif 1, motif 2, motif 3, motif 5, and motif 6 (Figure 3B). Furthermore, a substantially similar genetic organization is identified in the same evolutionary group of AaIAA genes, such as AaIAA22, AaIAA23, AaIAA55, and AaIAA56, indicating that duplicated genes may exert the same function. Moreover, four typical conserved domains, including domain I (motif 6), domain II (motif 3), domain III (motif 5), and domain IV (motif 1), were mapped (Figure 3C). A total of 46 AaIAA proteins possess four typical domains, whereas a subset of the IAA proteins possesses reduced numbers of domains, including motif 6 (domain I), which is missing in AaIAA2/7/18/20/25/26/29/34/47/52/54/57/61; AaIAA7 and AaIAA29 only have domain IV; AaIAA20, AaIAA52, AaIAA54, and AaIAA61 contain domain III and domain IV; AaIAA26 and AaIAA34 contain domain II and domain IV; and AaIAA31 is missing domain IV (Figure 3B). Notably, proteins in the same subset exhibit comparable distributions.

The 2000 bp regions of the *AaIAA* genes’ promoter sequences were interrogated to find *cis*-elements to determine the expression level responses of *AaIAA* genes to adversity or hormonal stresses. The findings exhibited that AaIAA genes contain various plant hormone response and stress response elements, encompassing auxin, abscisic acid (ABA), salicylic acid (SA), methyl jasmonate (MeJA) and gibberellin-responsive and drought-inducible cis-elements (Figure 3D). Regarding the auxin-responsive elements, 38 auxin-responsive elements, including 30 *AaIAA* genes, were identified. In particular, four auxin-responsive elements were found in the promoter regions of *AaIAA3*, indicating that *AaIAA3* has a strong function in the auxin-responsive domain. There are 41, 31, 34, and 43 *AaIAAs* genes containing MeJA-responsive, salicylic acid, gibberellin-responsive, and abscisic acid-responsive elements, respectively. Furthermore, various classes of hormone-associated *cis*-elements were identified in the promoter sequences of many AaIAA genes, including *AaIAA1*, *AaIAA3*, *AaIAA5*, and *AaIAA9*. The presence of *cis*-regulatory sequences exhibits a possible role of the Aux/IAA genes in crosstalk regulation by various hormones. Additionally, all *AaIAA* genes contain light-responsive elements varying from 5 to 21, and some genes contain tissue specificity *cis*-elements, such as *cis*-regulatory elements involved in endosperm expression in *AaIAA2*, *AaIAA9*, *AaIAA10*, *AaIAA23*, *AaIAA60*, *AaIAA30*, *AaIAA34*, *AaIAA36*, *AaIAA39*, *AaIAA47*, *AaIAA55*, and *AaIAA56*; *AaIAA18*, *AaIAA19*, *AaIAA25*, *AaIAA26*, *AaIAA34*, *AaIAA38*, and *AaIAA57* have *cis*-regulatory elements involved in flavonoid biosynthetic gene regulation; and *AaIAA20*, *AaIAA27*, *AaIAA28*, *AaIAA33*, and *AaIAA61* have *cis*-elements involved in differentiation of palisade mesophyll cells (Figure 3D).

### 2.4. GO Annotation of AaIAA Genes

Gene Ontology (GO) annotation was performed for all 61 Aux/IAA genes to examine the biological influences they exert (Appendix A). In biological processes, a maximum of 60 genes responded to the regulation of transcription, DNA-templated; followed by the auxin-activated signaling pathway (59 genes); response to auxin (56 genes); and transcription, DNA-templated (55 genes). In terms of cellular components, 59, 12, and 2 AaIAA genes were classified into the nucleus, intracellular, and integral components of the membrane, respectively. In addition, at the molecular level, 56 of 61 AaIAA unigenes were recognized as having DNA-binding transcription factor activity, while 48, 15, 11, and 7 AaIAA genes responded to protein binding, sequence-specific DNA binding, transcription regulatory region DNA binding, and DNA binding, respectively (Figure 4).

### 2.5. Phylogenetic Relationship of Aux/IAA Genes in A. thaliana and A. argyi

To examine the relationship of Aux/IAAs proteins between *A. argyi* and *A. thaliana*, phylogenetic analysis was performed using the predicted complete amino acid sequences of Aux/IAAs from *A. argyi* and *A. thaliana*. Each Aux/IAA protein was grouped into 13 distinct classes named Group 1 to 13 (Figure 5). *A. argyi* proteins were distributed across all groups except Group 3. In *A. thaliana*, Group 1 was restricted, and other groups except Group 7, 8, and 12 were expanded in *A. argyi*, including three Aux/IAA genes from *A. thaliana* and nine genes from *A. argyi* in Group 4 and five genes from *A. thaliana* and fifteen genes from *A. argyi* in Group 9. Orthologous genes typically have similar biological functions. There exist orthologous genes between *A. argyi* and *A. thaliana*, clustered into the same branch, including AaIAA34 to AtIAA33 in Group 1 and AaIAA5 and AaIAA37 to AtIAA6 and AtIAA39 in Group 7 (Figure 5). In addition, 21 pairs of proteins paralogous to sister pairs of AaIAA proteins in the same species that are clustered to the same branch were found, similar to the clustering result in Figure 3A, suggesting similar biological functions or functional redundancy.

### 2.6. Protein Interaction Prediction Analysis of Aux/IAA Proteins in A. argyi

To deeply comprehend the potential function of Aux/IAA proteins and the interaction between its family members in *A. argyi*, 61 AaIAA proteins were examined using STRING software (Version: 12.0) based on the homologous genes of *Artemisia annua* in the STRING database (Figure 6). There were 34 nodes and 145 edges, indicating that 34 proteins have 145 interactions, including 29 Aux/IAA proteins. Among them, AaIAA26 had the largest number of potentially interacting proteins and can potentially interact with 24 Aux/IAA members, followed by AaIAA29 which can potentially interact with 23 Aux/IAA members, and AaIAA32/36/38/4/41/44/47/50/51/54/57/59/61 proteins that had only one potential interacting Aux/IAA protein. These results suggest that AaIAA26/29/7/34 proteins are possibly core members of the IAA family interaction and potential functional collaborators between Aux/IAA proteins. Furthermore, for more protein interaction analysis, 26 and 25 Aux/IAA proteins can interact with the ARF proteins CTI12_AA579330 and CTI12_AA519960, respectively. AaIAA30, AaIAA47, and AaIAA61 proteins can potentially interact with Ubiquitin-related modifier 1 homolog proteins (CTI12_AA267880 and CTI12_AA498100), which operate as a sulfur carrier necessary for the 2-thiolation of mcm(5)S(2)U at the tRNA wobble positions of cytosolic tRNA(Lys), tRNA(Glu), and tRNA(Gln), acting as a sulfur donor in tRNA 2-thiolation reaction through thiocarboxylation (-COSH) at its C-terminus by MOCS3 [33]. Additionally, AaIAA7 proteins can potentially interact with the Mediator of RNA polymerase II transcription subunit 14 protein (CTI12_AA036660), which operates as a constituent of the Mediator complex, a coactivator acting to regulate the transcription of almost all RNA polymerase II-associated genes [34]. Therefore, this protein interaction network analysis offers evidence for further validation of the function of IAA proteins, supporting the idea that IAA proteins are involved in multiple biological processes through their interactions with target proteins.

### 2.7. Expression Analysis of AaIAA Genes in Leaf and Root, and in Response to Salt and Saline-Alkali Stresse

To comprehend the spatial pattern expression and the putative function of *IAA* genes in *A. argyi*, we analyzed their expression profiles base on fragments per kilobase of exon model per million mapped fragments (FPKM) values in leaves and roots using available RNA-seq datasets (NCBI accession: PRJNA1054765) (Figure 7A, Appendix A). Tissue expression analysis in roots and leaves through RNA-seq data exhibited that *AaIAA40*, *AaIAA41*, *AaIAA47*, *AaIAA50*, and *AaIAA54* genes were undetected in the leaf and root of RNA-seq data, indicating that these genes may be found in a distinct tissue-specific expression pattern in other tissues or organs, such as stems. *AaIAA2*, *AaIAA17*, *AaIAA20*, *AaIAA30*, *AaIAA34*, *AaIAA37*, *AaIAA52*, and *AaIAA61* genes were lowly expressed, or extremely lowly expressed, or had no expression in leaves or roots. The remaining 48 *AaIAA* genes were used for analysis, and a heatmap was constructed (Figure 7A). The results demonstrated that the expression levels of *AaIAA5*, *AaIAA11*, *AaIAA19*, *AaIAA51*, and *AaIAA58* genes were significantly higher in leaves than in roots. Most remaining members were significantly higher in roots than in leaves, including *AaIAA1*/*4*/*8*/*10*/*13*/*14*/*15*/*18*/*22*/*23*/*24*/*27*/*28*/*29*/*35*/*36*/*38*/*39*/*42*/*43*/*44*/*45*/*46*/*48*/*49*/*53*/*55*/*56*/*57* genes (Figure 7A). In addition, low abundance expression analysis exhibited some members with average FPKM < 1 with low expression levels in leaves [35], including *AaIAA4*/*14*/*22*/*38*/*45*/*55*/*56*. Only *AaIAA5* exhibited low expression levels in roots among the remaining 48 members. These results revealed that *AaIAA* genes exhibited tissue-specific expression patterns in *A. argyi*.

Furthermore, to comprehend the expression patterns of *Aux*/*IAA* genes under salt and saline–alkali treatment using available RNA-seq data (NCBI accession: PRJNA1054765) in *A. argyi*, *Aux*/*IAA* gene expression in salt and saline–alkali stress RNA-seq data were analyzed (Appendix A). Members with FPKM value = 0 in any set were excluded, and the remaining 46 *AaIAA* genes were used for analysis. The results showed that most Aux/IAA members responded to salt and saline–alkali stresses (Figure 7B,C). In leaves, more AaIAA members exhibited reductions under salt and saline–alkali stresses. Among them, twelve genes were significantly responsive to salt stress, including nine genes (*AaIAA5*/*11*/*12*/*18*/*48*/*49*/*57*/*58*/*59*) downregulated and two genes (*AaIAA8* and *AaIAA35*) upregulated; sixteen genes significantly responded to saline–alkali stress, including twelve genes (*AaIAA5*/*11*/*12*/*13*/*15*/*16*/*18*/*24*/*58*/*59*/*44*/*48*/*49*/*57*) downregulated and two genes (*AaIAA8* and *AaIAA35*) upregulated (Figure 7B). In roots, most of the *AaIAA* genes exhibited a decrease under salt and saline–alkali stresses. Among them, three genes (*AaIAA1*, *AaIAA27*, and *AaIAA28*) significantly responded to salt stress in downregulated form, and six genes (*AaIAA1*/*24*/*26*/*27*/*28*/*29*) exhibited significant downregulation in response to saline–alkali stress (Figure 7C). Additionally, some *AaIAA* genes exhibited opposite expression patterns in response to salt and saline–alkali stresses, such as *AaIAA35*, *AaIAA8*, *AaIAA27*, *AaIAA1*, and *AaIAA28*, showing increases in leaves and decreases in roots (Figure 7B,C). Generally, most *Aux*/*IAA* genes exhibited similar expression patterns in response to salt stress and saline–alkali stress, and the expression of most *Aux*/*IAA* genes was decreased under salt and saline–alkali stresses, indicating that the growth of *A. argyi* was inhibited under salt and saline-alkali stresses.

### 2.8. Differential Expression Patterns of AaAux/IAA Genes in Response to Different Hormones Treatment

Auxin represents a critical growth modulator controlling plant growth, recruiting the assistance of various endogenous signals, encompassing various plant hormones like abscisic acid, gibberellic acid, jasmonates, and salicylic acid [36]. The Aux/IAA gene family plays a regulatory role in auxin regulating plant growth, and it is also involved in the response to other hormones. To assess the physiological functions of *AaIAA* genes and their reactions to auxin and different hormones, 12 genes, with FPKM > 10 and no sister pair genes, were chosen to examine their expression under indole-3-acetic acid (IAA), salicylic acid (SA), methyl jasmonate (MeJA), and abscisic acid (ABA) treatments, and the results of quantitative real time polymerase chain reaction (RT-qPCR) showing the expression of 12 *AaIAA* genes responses to the hormones of IAA, SA, MeJA, and ABA (Figure 8). Under IAA treatment, 11 members exhibited significantly increased expression, especially *AaIAA13* after 3 h and *AaIAA12*/*21*/*33*/*35* after 12 h. While *AaIAA19* exhibited decreased expression, especially after 12 h, *AaIAA7*/*9*/*26*/*36* also showed reduced expression after 3 h (Figure 8A). Under ABA treatment, *AaIAA21*/*26*/*33*/*35*/*36* exhibited increased expression; *AaIAA7*/*9*/*11*/*12*/*13*/*19*/*57* exhibited both upregulated and downregulated expression at different time points of ABA treatment, such as the expression of *AaIAA7* exhibiting significantly increased expression after 12 h, while it decreased after 3 h and 24 h. Among them, *AaIAA35* exhibited significant upregulation (up to 14.48-fold). The *AaIAA35* gene plays a core role in the response to ABA of the Aux/IAA family in *A. argyi* (Figure 8B). Under SA treatment, *AaIAA21*/*36* showed increased expression; *AaIAA26*/*33*/*35*/*57* showed decreased expression; and the remaining six *AaIAAs* genes showed both upregulated and downregulated expression at different time points of SA treatment, including the expression of *AaIAA7* showing significant decreases after 3 h and 12 h, while it increased after 24 h and 48 h. Among them, *AaIAA11* and *AaIAA12* were significantly up-regulated (up to 4.24-fold). The *AaIAA11* and *AaIAA12* genes play core roles in response to SA of the Aux/IAA family in *A. argyi* (Figure 8C). Under MeJA treatment, *AaIAA7*/*13*/*19* showed decreased expression; *AaIAA21*/*33*/*35*/*36* showed increased expression; *AaIAA9*/*11*/*12*/*26*/*57* showed both upregulated and downregulated expression at different time points of MeJA treatment, such as the expression of *AaIAA9* showing significant increases after 3 h and 12 h, while it was decreased after 24 h and 48 h. Among them, *AaIAA35* exhibited significant upregulation (up to 33.27-fold). The *AaIAA35* gene plays a core role in response to ABA of the Aux/IAA family in *A. argyi* (Figure 8D).

## 3. Discussion

Recently, with the continuous examination of the genomes of medicinal plants, genome sequencing research has enabled an opportunity to extract gene classes through a genome-wide test. The analysis of the *A. argyi* genome provides important genetic resources for the mining and functional analysis of gene families. Auxin signaling acts as a critical signaling avenue throughout various plant biological processes, including development, organogenesis, and response to various environmental alterations [37]. Aux/IAA genes modulate auxin-triggered gene expression and varied aspects of plant growth through the Aux/IAA–ARF and Aux/IAA–TIR complexes [37,38]. Consequently, research into the function of the Aux/IAA gene family in *A. argyi* is beneficial for analyzing *A. argyi* growth and development, stress resistance, and other biological processes. In our study, the systematic characterization of the *A. argyi* Aux/IAA gene class members outlined offer novel insights into the possible roles of various Aux/IAA genes in controlling plant reactions to auxin and their potential functions.

### 3.1. Characterization and Structure of the AaIAA Family Genes in A. argyi

Many Aux/IAA genes that regulate auxin signal transduction and auxin degradation have been identified in various plants by the application of physiological, genetic, molecular, and biochemical methods, primarily through complete genomic sequencing. In this study, 61 AaIAA genes were identified, and the number of AaIAA members in *A. argyi* was much higher than in tomato (26 members), *A. thaliana* (29 members), rice (31 members), and maize (31 members), and similar in number to diploid *Brassica rapa* (55 members) [39], polyploid *Brassica oleracea* (60 members), and the paleopolyploid soybean (63 members) [40]. *A. argyi* is an allotetraploid with 34 chromosomes and 34 pseudochromosomes, separated into two haplotypes based on self-comparison. The longer and more complete chromosomes were assigned to haplotype group A, which contained pseudochromosomes 1 to 17 and was used for our genomic analysis [30]. Therefore, 61 AaAux/IAA genes identified in our study might represent half the members of *A. argyi*, similar to the same allotetraploid *Brassica napus* in Aux/IAA genes (119 members) [41]. In our study, compared to *A. argyi* and *A. thaliana*, gene divergence and duplication events showed that 35 Aux/IAA genes in *A. argyi* had orthologs with 21 members of *A. thaliana*. Most of the *A. argyi* Aux/IAA genes had one (13 genes) or two (13 genes) orthologs in the *A. thaliana* genome, seven *A. argyi* Aux/IAA genes each had three orthologs in *A. thaliana*, and another two *A. argyi* Aux/IAA genes had four orthologs in *A. thaliana*. Theoretically, there should be four ortholog genes for *A. thaliana* Aux/IAA genes in *A. argyi*, implying that most Aux/IAA genes experienced gene loss in the tandem duplication process of the Aux/IAA gene family in *A. argyi*. Moreover, the phylogenetic relationship of Aux/IAA genes analysis showed that Aux/IAA genes mostly exist as sister pairs of genes, suggesting there were gene duplication events, consistent with the whole gene duplication event that *A. argyi* underwent after divergence from *Artemisia annua* [30].

Previous studies have demonstrated that Aux/IAA genes are functionally different. The differentiation of the Aux/IAA genes in *A. thaliana* may depend on both the molecular structure of proteins and expression patterns. In our study, 46 members possess the four typical domains, I, II, III, and IV. A total of 15 non-canonical Aux/IAA proteins were found to lack single or multiple domains, contributing to their divergence. Among them, domain I contains conserved leucine residues and the LXLXLX motif. The LXLXLX motif plays a vital role in the repression of IAA proteins [42]. Mutation at any of these three Leu residues in this motif results in total loss of repression or strongly reduced repression in the case of the mutation in the third Leu in the motif [42]. In this study, AaIAA2/7/18/20/25/26/29/34/47/52/54/57/61 proteins do not contain domain I, implying that these proteins could not act as a repressor in auxin signaling and may function differently. Eight Aux/IAA proteins (AaIAA2/7/26/29/30/34/54) do not contain Domain II, consistent with evidence from the AtIAA20 lacking domain II, and cannot be rapidly degraded in the presence of basal or increased levels of auxin in *A. thaliana*, suggesting that these genes can repress auxin response gene expression [43]. Domains III and IV can bind to the ARF, and are responsible for homo- and hetero-dimerization between Aux/IAA and ARF proteins [38]. Five Aux/IAA proteins (AaIAA5/7/26/29/34) do not contain domain III, suggesting that these genes cannot bind ARF or function independently. In addition, most of the AaIAA proteins contain domains III–IV, which are important for binding to ARFs. AaIAA proteins usually directly bind to ARF proteins and regulate the expression of downstream auxin response genes [44].

### 3.2. Tissue-Specific Expression of Aux/IAA Genes in A. argyi

Tissue-specific and developmental stage-specific expression of Aux/IAA genes has been established across various species. In *Hedychium coronarium*, the tissue-preferential and stage-specific expressions exhibited several Aux/IAA genes expressed in flowers, suggesting their involvement in the biology of specific tissues and flower scent formation [45]. In *Triticum aestivum*, *TaIAA27* and *TaIAA31* exhibit tissue-specific expression in roots and seeds [46]. In *Brassica napus*, the RNA-seq data indicated that the expression of Aux/IAA members was highly variable among tissues, and 24 Aux/IAA genes had tissue(s)-specific expression patterns [41]. In our study, tissue expression analysis in roots and leaves in RNA-seq data demonstrated that the expression level of AaIAAs differed, such as *AaIAA11*, *AaIAA19*, *AaIAA20*, *AaIAA30*, *AaIAA51*, *AaIAA58*, and *AaIAA61* genes being significantly higher in leaves than in roots, and 25 remaining members exhibited significantly higher levels in roots than that in leaves. These results indicated that more AaIAA members act on the roots and regulate plant growth and development, similar to findings in *Fagopyrum tataricum*, which also exhibited significantly lower expression levels of most *FtAux*/*IAA* genes in the leaves than in root tissues [47]. Previous studies have shown that Aux/IAA genes are involved in root growth and development, and *OsIAA11* is expressed in the root tip, lateral root, middle column, and lateral root primordia [48]. In *A. thaliana*, *AtIAA33* plays a crucial role in maintaining apical stem cell activity [49]. In addition, *AaIAA40*, *AaIAA41*, *AaIAA47*, *AaIAA50*, and *AaIAA54* genes were not detected in leaf and root of RNA-seq data, indicating that these genes may illustrate a distinct tissue-specific expression pattern in other tissues in *A. argyi*, and tissue-specific expression of these members requires further analysis.

### 3.3. Adversity and Hormone Responses of Aux/IAA Genes in A. argyi

Gene expression patterns in *A. argyi* responses to salt and saline–alkali stresses and different hormone stimuli were identified using RNA-seq data or RT-qPCR analysis, delivering novel insights into the potential functions in governing plant growth and development by *AaIAA* genes. Aux/IAA is critical, acting as a class of primary auxin-responsive genes, rapidly induced by auxin. In *Fagopyrum tataricum*, all Aux/IAA genes exhibited an intricate response pathway under IAA treatment; among them, different genes exhibited differing trends upon treatment for extended durations, and the expression of some genes differed in different tissues [47]. In our study, 12 *AaIAA* genes also showed a complex pattern in response to IAA treatment by RT-qPCR. Although only two members of the selected 12 members contained auxin-responsive *cis*-elements in their promoters, the IAA family acted as a class of primary auxin-responsive genes, involved in auxin responses. Furthermore, the Aux/IAA family regulates different abiotic stresses. The level of glucosinolates (GLS) is regulated by *IAA5*, *IAA6*, and *IAA19* in the auxin Aux/IAA protein family, and *AtIAA5*/*6*/*19* deficiency causes decreased GLS levels and decreased drought tolerance, indicating that Aux/IAA proteins regulate drought tolerance in *Arabidopsis* by modulating glucosinolate levels [19]. In rice, some Aux/IAA genes were regulated by abiotic stresses such as drought, salt, and low temperature, such as *OsIAA9* and *OsIAA20* being significantly upregulated under high-salt conditions [20]. In our study, RNA-seq data analysis exhibited that 12 and 16 *AaIAA* genes were significantly responsive to salt stress and saline–alkali stress in leaves, respectively. Most *AaIAA* genes exhibited decreased expression under salt and saline–alkali stresses in roots, indicating that the growth of *A. argyi* was inhibited under salt and saline–alkali stresses. These findings indicated that the Aux/IAA family regulates the response to salt and saline–alkali stresses in *A. argyi*.

Additionally, auxin is a critical growth regulator that governs plant development, and it is recruited by other endogenous signals, including abscisic acid, brassinosteroids, cytokinins, ethylene, gibberellic acid, jasmonates, and salicylic acid [36]. As an essential gene family that responds to auxin signaling, it is speculated that other hormones regulate the Aux/IAA gene family. Prior studies have focused on the Aux/IAA family response to ethylene regulation. The downregulation of *SlIAA3* results in auxin- and ethylene-related developmental defects, supporting the hypothesis that *SlIAA3* is a molecular link between ethylene and auxin signaling in tomatoes [50]. The Aux/IAA gene family also mediates the interaction between auxin and the signaling of other hormones, such as abscisic acid, cytokinin, and salicylic acid. *IAA16* confers reduced responses to auxin and abscisic acid and impedes plant growth and fertility [51]. In cotton, the enhanced resistance in *GhIAA43*-silenced cotton plants is because of the activation of SA-related defenses, suggesting that *GhIAA43* can act as a molecular link between SA and auxin signaling [52]. In a study of tobacco mosaic virus-directed reprogramming of Aux/IAA proteins, transcriptional expression studies demonstrate a role for Aux/IAA-interacting proteins in the regulation of SA and JA host defense responses, as well as virus-specific movement factors [53]. In our study, 12 *AaIAA* genes can respond to ABA, SA, and MeJA treatments, but their expression levels varied considerably. For ABA responsiveness, 8 of 12 *AaIAA* selected genes except *AaIAA11*/*13*/*21*/*35* contained a *cis*-acting element involved in abscisic acid responsiveness. For MeJA responsiveness, 9 of 12 *AaIAA* selected genes except for *AaIAA11*/*26*/*57* contained *cis*-acting regulatory elements involved in MeJA responsiveness. Only five members, including AaIAA9/11/13/35/36, for SA responsiveness, contain cis-acting elements involved in salicylic acid responsiveness. RT-qPCR results demonstrated that some members could respond to related hormones. However, they did not contain corresponding regulatory *cis*-elements in their promoters, and there may be other regulatory modes responding to hormone regulation, suggesting that *AaIAA* genes may exert an essential role in regulating plant growth and development by auxin interactions with other hormones. In addition, we know that hormone concentrations influence the response patterns of different genes to hormones. For example, high concentrations cause non-specific effects and cannot be considered as only specific hormones. Thus, the next study should develop a dose-response curve to provide a more accurate regulatory model for AaIAA family members to respond to different hormone regulation. Overall, the functional characterization and expression analysis of Aux/IAA family genes assist in revealing the mechanisms underlying how auxin signaling is involved in plant growth, responses to environmental changes, and interactions with other hormones in a spatiotemporal-specific manner in *A. argyi*.

### 3.4. Possible Correlation between Aux/IAA Gene and Switching to Secondary Metabolism in A. argyi

Previous studies have shown that various stresses and hormones can regulate the accumulation of secondary metabolites, which in turn controls the quality of medicinal plants [54,55]. In *Glycyrrhiza uralensis*, exogenous ABA has the ability to stimulate the synthesis of Glycyrrhiza uralensis’s active components, such as isoliquiritin, glycyrrhizin, liquiritigenin, and isoliquiritin [54]. In *Ruta graveolens*, salicylic acid can promote the accumulation of phenols and flavonoids in callus [55]. In cultured *Onosma paniculatum* cells, brassinolide together with IAA (indoleacetic acid) and BAP (6-benzylaminopurine) can promote the accumulation of shikonin at appropriate concentrations [56]. In apple, it has been reported that auxin has the ability to regulate anthocyanin biosynthesis via the Aux/IAA-ARF signaling pathway [57]. In *A. argyi*, previous studies have shown that exogenous hormone (MeJA, SA and ABA) treatments changed the eight secondary metabolites, including phenylpropanoids, flavonoids, terpenoids, alkaloids, and others [58]. In our study, the expression levels of 12 *AaIAA* genes changed to varying degrees in response to IAA, SA, MeJA, and ABA, and various hormones had an impact on the accumulation of secondary metabolites in *A. argyi*. it is speculated that MeJA, SA, IAA, and ABA may have regulated the accumulation of secondary metabolites in *A. argyi* through the Aux/IAA genes or Aux/IAA-related signaling pathway. In addition, Aux/IAA proteins, which are a crucial part of the auxin signaling system, are essential for controlling the growth process of plants [5]. Plant secondary metabolite accumulation is a dynamic process that is ever-changing during the growth phase of the plant [59]. It is suggested that AaIAA proteins may present as a potential mediator of plant development and secondary metabolite accumulation. Above all, more research is required to understand how the Aux/IAA protein in *A. argyi* regulates the accumulation of secondary metabolites.

## 4. Materials and Methods

### 4.1. Plant Material, Growth Conditions, and Hormone Treatment

The plant materials used in our study were obtained from the medicinal botanical garden of Henan University of Chinese Medicine, Zhengzhou, Henan Province, China. The root buds of *A. argyi* were used as explants for disinfected, and sterile tissue culture seedlings of *A. argyi* were obtained as described by Li et al. [60].

Aseptic tissue culture seedlings of *A. argyi* plants were grown in tissue culture bottles with ½ Murashige and Skoog (½MS) culture medium containing 0.7% agar and 3% sucrose [60]. The medium has a high concentration of inorganic salts and ions, also includes iodine and chloride, is a relatively stable ion balance solution, has a high nitrate content, and can meet the nutritional and physiological needs of most plant cells [61]. The seedlings were grown in sterile conditions under a 16 h/8 h light/dark photoperiod (120 µmol m^−2^ s^−1^) at 24 to 26 °C. After 1 month of growth, the *A. argyi* seedlings were treated with different hormones. For the salicylic acid (SA), methyl jasmonate (MeJA), and abscisic acid (ABA) treatments, 200 µmol SA, 200 µmol MeJA, and 200 µmol ABA were evenly sprayed on the leaves of *A. argyi* after uncapping the bottle, then the bottle was capped after the spraying was finished. The control group was sprayed with sterile water. For the indole-3-acetic acid (IAA) treatment, two sets of *A. argyi* seedlings were incubated in liquid ½MS medium with and without 10 μM IAA for 0 h, 3 h, 12 h, 24 h, and 48 h. The leaves of all samples between three and seven nodes were harvested at 0 h, 3 h, 12 h, 24 h, and 48 h following treatment. All collected samples were frozen in liquid nitrogen immediately and preserved at −80 °C for subsequent RT-qPCR analysis.

For RNA-seq analysis, the 3-month-old *A. argyi* seedlings that were transplanted from tissue culture seedlings and grown in an illumination incubator with a relative humidity of 70% during a 16 h/8 h light/dark photoperiod (120 µmol m^−2^ s^−1^) at 24 to 26 °C were selected for stress treatments. The uniformly developed seedlings of *A. argyi* were fully watered using a 200 mM total solution (NaCl/Na_2_SO_4_ = 1:1, pH = 7.0) for the salt stress treatment, 200 mM total (NaCl/Na_2_SO_4_/NaHCO_3_/Na_2_CO_3_ = 1:1:1:1, pH = 9.0) for the saline–alkali stress treatment, and water treatment as a control. After three days of treatment, the leaves from four to seven nodes were removed, and as soon as the roots were thoroughly cleaned in water, samples were taken. Upon collection, every sample was promptly frozen in liquid nitrogen and stored at −80 °C for later use.

### 4.2. Identification of Aux/IAA Genes in A. argyi

*A. argyi* genome annotations were obtained from the National Genomics Data Center (NGDC, https://ngdc.cncb.ac.cn; accessed on 10 November 2022) with the BioProject accession number PRJCA010808 [30]. HMMER and BLASTP were applied to search for *A. argyi* Aux/IAA proteins using the published *Arabidopsis* Aux/IAA protein sequences as the initial protein query. A Hidden Markov Model of Pfam3 tools was used to examine the Aux/IAA (PF02309) domain for all the obtained protein sequences. Physiochemical parameters of each gene were determined using ExPASy (https://web.expasy.org/protparam/; accessed on 12 March 2023). Information about genomic sequences, cDNA sequences, protein sequences, and feature information was obtained from the GFF file available on the *A. argyi* genome project webpages [30].

### 4.3. Gene Structural, Motif Scanning and Phylogenetic Analysis of Aux/IAA Genes in A. argyi

All AaAux/IAA genes were mapped to the chromosomes from the physical positional information obtained from the *A. argyi* genomic GFF files downloaded from the NGDC database using Tbtools [62]. The exon–intron organization of the AaIAA genes was examined using the Gene Structure Display Server (GSDS). The IrTIFY protein conserved domains were analyzed by Pfam (https://pfam.xfam.org; accessed on 15 March 2023), and the MEME website (https://meme-suite.org/meme/; accessed on 15 March 2023) was used to predict protein conserved motifs with the motif value established at 10. Multiple sequence alignments were conducted on the AaIAA protein sequences using ClustalW with default parameters. A phylogenetic tree was constructed using the aligned AaIAA protein sequences using MEGA (Version: 6.06) [63] and selecting the neighbor-joining method, with 1000 bootstrap replicates. The same methods were applied to analyze the evolutionary relationships between *A. argyi* and *Arabidopsis*. TBtools was employed to visualize the conserved domain, motif, and IrTIFY evolutionary tree.

### 4.4. Cis-Elements Analysis in the Promoter Regions of Aux/IAA Genes

To characterize *cis*-elements in the promoter region of each Aux/IAA gene in *A. argyi*, 2000 bp of genomic sequence upstream of the translation start site was chosen from the available *A. argyi* genome sequence using TBtools, and analyzed with PlantCARE (http://bioinformatics.psb.ugent.be/webtools/plantcare/html/; accessed on 25 March 2023). A total of 21 *cis*-elements were selected for analysis, including hormone-related *cis*-elements and stress-response *cis*-elements. Different colors represent different *cis*-elements, and *cis*-elements distribution was visualized using TBtools [62].

### 4.5. Collinearity and Specific Duplication Events Analysis

Multiple collinear scanning toolkits (MCScanX) were employed to analyze gene duplication events with default parameters [64]. To uncover the synteny relationship of orthologous Aux/IAA genes between *A. argyi* and *Arabidopsis*, syntenic analysis maps were constructed using the Dual Synteny Plotter software (available online: https://github.com/CJ-Chen/TBtools; accessed on 15 June 2023) [61]. Furthermore, the visualization of sequence similarity of the AaIAA genes in *A. argyi* was analyzed by Circoletto [32].

### 4.6. GO Functional Identification

The Gene Ontology (GO) terms for functional annotation were analyzed using Blast2GO software (https://www.blast2go.com/; accessed on 6 April 2023) with default parameters. The annotations of GO terms were investigated using the Gene Ontology Consortium (http://geneontology.org/; accessed on 6 April 2023).

### 4.7. Transcriptome Analysis Based on RNA-Seq Data

To uncover the expression profile of Aux/IAA genes in various tissues in response to salt and saline–alkali stress in *A. argyi*, RNA-seq data from our study, including leaves and roots with or without salt and saline–alkali stress for three days, were used for analysis (NCBI accession: PRJNA1054765). Fragments per kilobase per million mapped reads (FPKM) were calculated using the Cufflinks tool (v2.2.1) to estimate gene expression levels. Heatmaps were constructed using the OmicStudio tools at https://www.omicstudio.cn/tool; accessed on 10 December 2023 [65] based on log_2_(FPKM values), data were normalized using Z-score, and each set underwent three replicates.

### 4.8. RNA Extraction and RT-qPCR Analysis

Total RNA from different samples of *A. argyi* was extracted using the EASYspin Plus Plant RNA Kit (AidLab, Beijing, China) following the manufacturer’s instructions. A sample of 1 µg of quantified RNA samples was prepared for the synthesis of cDNA strands using the EasyScript All-in-One First-Strand cDNA Synthesis SuperMix for qPCR (One-Step gDNA Removal) (Transgen, Beijing, China) in accordance with the manufacturer’s instructions. The RT-qPCR experimental steps were performed according to a previously published protocol [66]. RT-qPCR amplification was conducted using TransStart Top Green qPCR SuperMix (Transgen, Beijing, China) according to the manufacturer’s instructions and was performed on the ABI QuantStudio 5 (Thermofisher Scientific, Waltham, MA, USA). Each RT-qPCR reaction contained 20 µL, including 10 µL 2× TransStart Top Green qPCR SuperMix, 1 µL cDNA template, 0.4 µL gene-specific primers (10 µM), 0.4 µL 50× Passive Reference Dye II, and 7.8 µL Nuclease-free water. The reaction conditions were 94 °C for 30 s, 45 cycles of 94 °C for 5 s, and 60 °C for 30 s. Each assay included three biological replicates and three technical replicates. *Actin* was used as an internal reference gene. The relative expression level was determined using the 2^−ΔΔCt^ method [67]. All primers are listed in Appendix A.

## 5. Conclusions

In this study, a genome-wide analysis of the Aux/IAA gene family in *A. argyi* was conducted, and 61 Aux/IAA genes were identified in the *A. argyi* genome by bioinformatic analysis. Their physical and chemical characters, chromosome locations, gene structure, motif scanning, GO annotation, promoter *cis*-elements, phylogenetic relationships, and protein interaction were examined. Furthermore, Aux/IAA gene expression patterns in roots and leaves, salt, and saline-alkali stresses were examined by RNA-seq data, and the expression patterns in response to hormones, including IAA, ABA, SA, and MeJA, were also confirmed through RT-qPCR, exhibiting a complex response pattern, suggesting that Aux/IAA may have a role in connecting auxin to other hormones. Overall, our study analyzed the Aux/IAA family of *A. argyi* and its expression patterns. Our data will act as a significant resource for further investigations on functional genomics of the Aux/IAA genes and expand the possibilities for studying plant growth and development in medicinal plant breeding.

## Figures and Tables

**Figure 1 plants-13-00564-f001:**
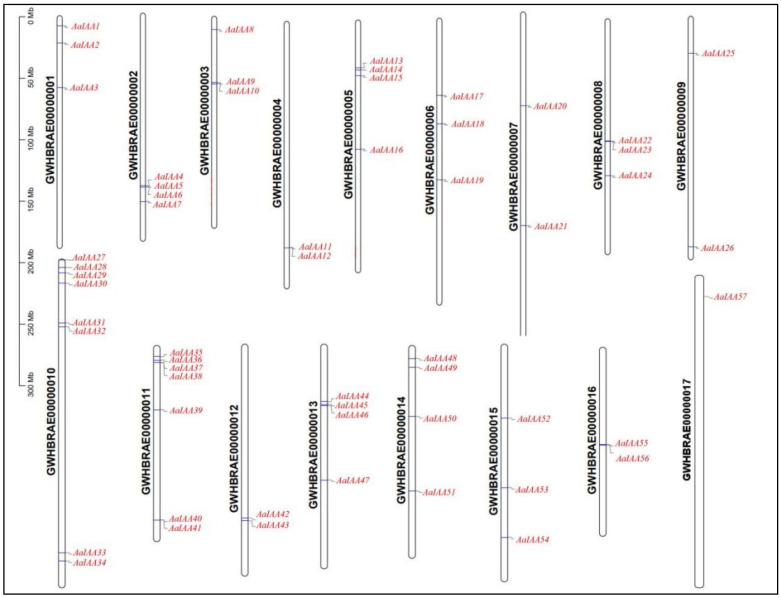
Chromosome distribution mapping of Aux/IAA gene family in *A. argyi*.

**Figure 2 plants-13-00564-f002:**
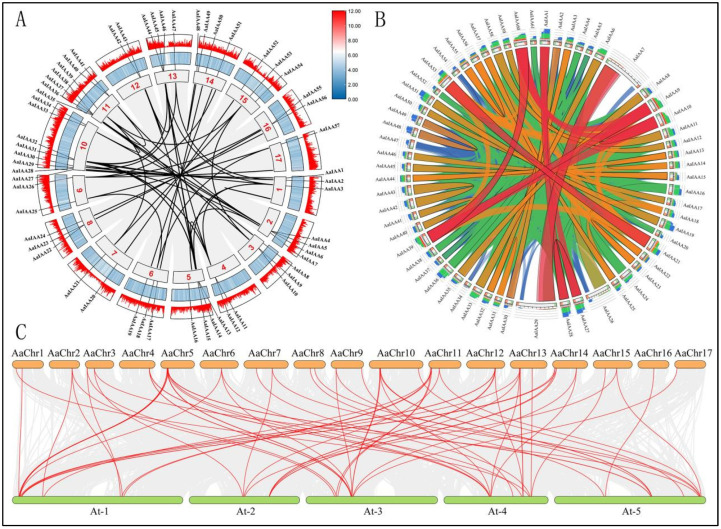
Collinear prediction and sequence similarity analysis of the Aux/IAA genes in *A. argyi*. (**A**): Mapping of collinear prediction of the AaIAA family genes in *A. argyi*. Black lines show AaIAA paralogs, red maths indicates chromosome numbers. (**B**): Visualising sequence similarity of the AaIAA genes in *A. argyi*. Different colored ribbons represent different levels of similarity, blue ≤ 50%, green ≤ 75%, orange ≤ 99%, red > 99%. (**C**): Collinear prediction of the Aux/IAA family genes between *A. thaliana* and *A. argyi*. Red lines show the collinear events.

**Figure 3 plants-13-00564-f003:**
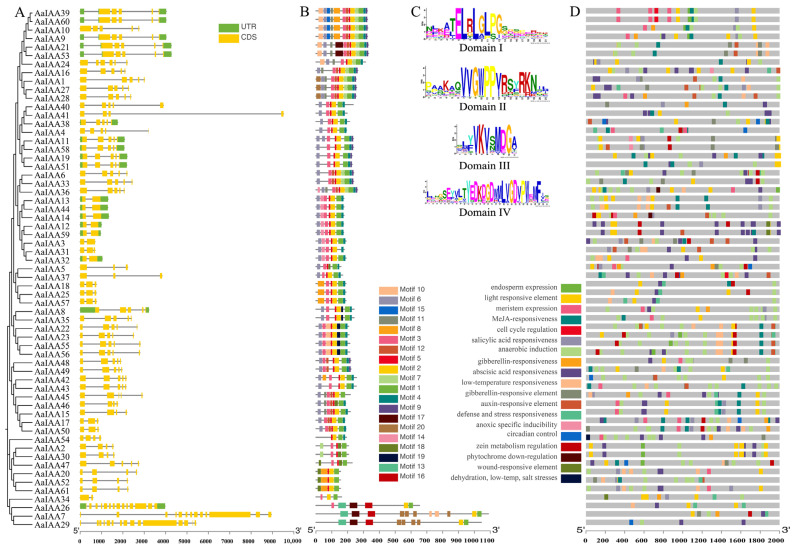
Gene structure, conserved motif, conserved domains, and promoter *cis*-element analysis of the Aux/IAA family genes in *A. argyi*. (**A**) Exon–intron organization of *A. argyi* Aux/IAA genes. (**B**) Conserved motif analysis of *A. argyi* Aux/IAA proteins. (**C**) The amino acid sequence of four typical conserved domains. (**D**) *Cis*-element analysis in the promoters of *A. argyi* Aux/IAA genes.

**Figure 4 plants-13-00564-f004:**
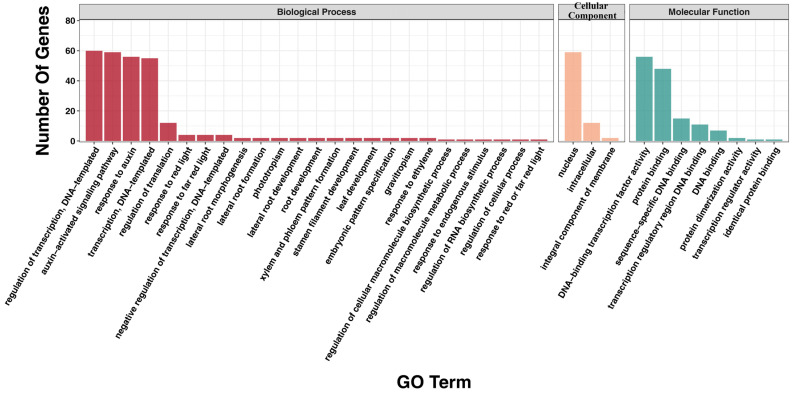
GO annotation of the Aux/IAA gene family in *A. argyi*.

**Figure 5 plants-13-00564-f005:**
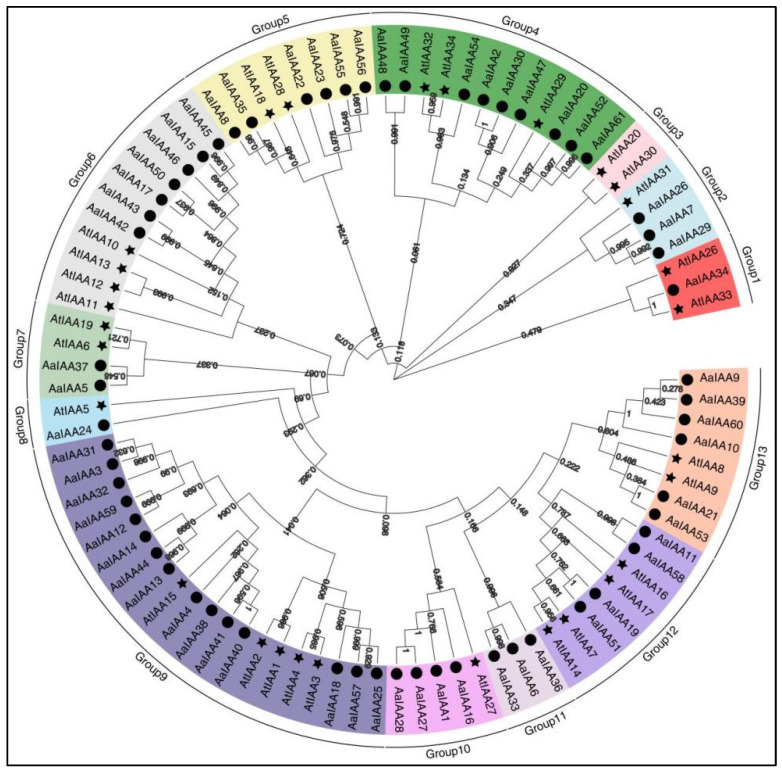
Phylogenetic investigation of Aux/IAA proteins in *A. argyi* and *A. thaliana*. The phylogenetic tree was developed using MEGA software (Version: 6.06) and the neighbor-joining method with 1000 bootstraps. Each Aux/IAA group (1 to 13) is indicated by a specific color. AaIAAs in *A. argyi* are indicated by a solid circle dot, and AtIAAs in *Arabidopsis* are indicated by a star.

**Figure 6 plants-13-00564-f006:**
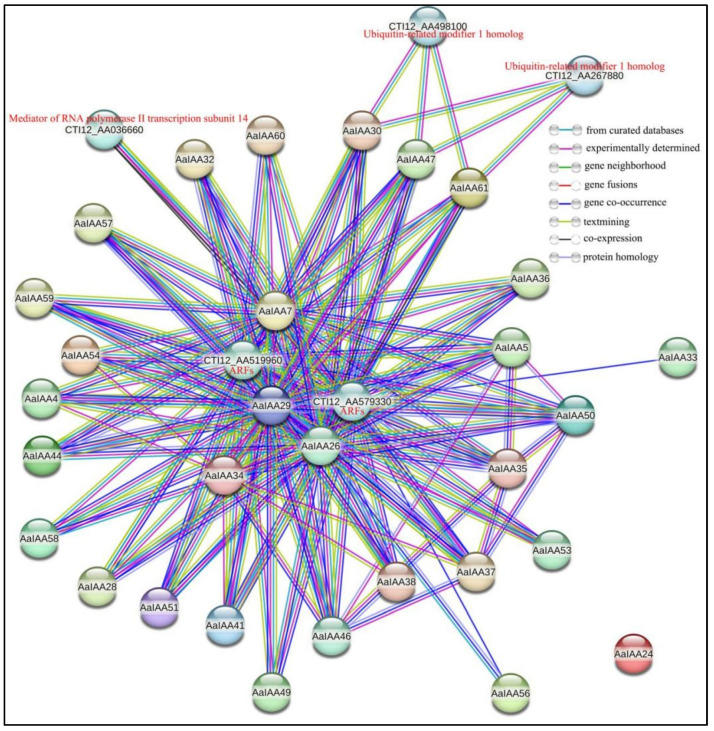
Prediction analysis of Aux/IAA protein interaction in *A. argyi*. Different line colors indicate the type of interaction evidence.

**Figure 7 plants-13-00564-f007:**
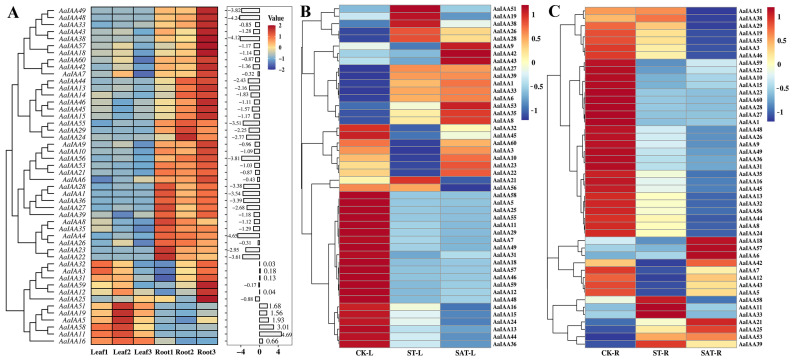
Expression profiles of AaIAA genes in the leaves and roots of *A. argyi* in response to salt and saline-alkali stresses based on RNA-seq data. (**A**) The expression profiles of *AaIAA* genes in the leaves and roots of *A. argyi*; FPKM data were normalized by Z-score; the bar chart shows the log_2_FC values of root vs. leaf. (**B**) Expression profiles of AaIAA genes in response to salt and saline–alkali stresses in leaves. (**C**) Expression profiles of AaIAA genes in response to salt and saline–alkali stresses in roots. The log_2_(FPKM values) data were normalized by Z-score. R: root, L: leaf, CK: control group, ST: salt treatment, SAT: saline–alkali treatment. Each set had three replicates.

**Figure 8 plants-13-00564-f008:**
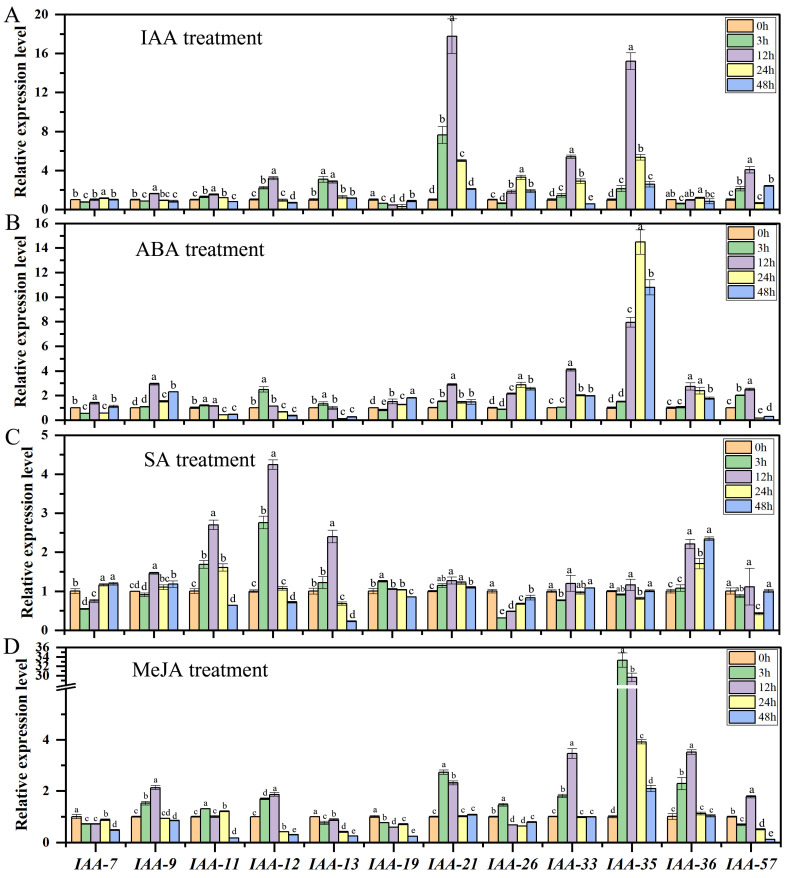
Expression profiles of AaIAA genes under various hormonal treatments. (**A**) IAA treatment. (**B**) ABA treatment. (**C**) SA treatment. (**D**) MeJA treatment. Error bars represent standard error of the mean; data shown are means ± SE. Letters “a,b,c…” indicate significant differences at *p* < 0.05.

**Table 1 plants-13-00564-t001:** Physical and chemical characters of AaIAAs proteins in *A. argyi.*

Gene Name	Gene ID	Amino Acids Number (aa)	Molecular Weight (Da)	Isoelectric Point (pI)	Instability Index	Aliphatic Index	Grand Average of Hydropathicity	Subcellular Localization
*AaIAA1*	Aarg01G001560.1	255	28,351.05	8.37	50.14	58.9	−0.687	Nuclear
*AaIAA2*	Aarg01G004030.1	206	23,404.46	6.88	47.62	69.95	−0.702	Nuclear
*AaIAA3*	Aarg01G011540.1	191	21,670.62	5.62	60.92	63.25	−0.699	Nuclear
*AaIAA4*	Aarg02G020290.1	194	22,104.6	8.98	40.02	81.24	−0.424	Nuclear
*AaIAA5*	Aarg02G020300.1	158	17,806.49	5.64	40.72	65.95	−0.477	Nuclear
*AaIAA6*	Aarg02G020570.1	239	26,431.46	8.59	30.31	74.18	−0.562	Nuclear
*AaIAA7*	Aarg02G023990.1	1089	121,872.7	6.28	62.05	69.94	−0.642	Nuclear
*AaIAA8*	Aarg03G002350.1	241	26,448.04	9.18	42.86	75.56	−0.637	Nuclear
*AaIAA9*	Aarg03G012510.1	326	34,600.87	7.53	42.41	71.81	−0.495	Nuclear
*AaIAA10*	Aarg03G012760.1	328	34,784.98	7.55	42.21	69.88	−0.525	Nuclear
*AaIAA11*	Aarg04G026560.1	237	26,131.52	8.29	39.93	60.89	−0.691	Nuclear
*AaIAA12*	Aarg04G026570.1	175	19,816.62	5.39	64.26	71.83	−0.574	Nuclear
*AaIAA13*	Aarg05G010530.1	177	20,054.95	7.66	57.36	66.05	−0.685	Nuclear
*AaIAA14*	Aarg05G010900.1	177	20,024.97	8.39	55.97	67.12	−0.653	Nuclear
*AaIAA15*	Aarg05G011700.1	217	23,924.21	7	50.43	79.4	−0.464	Nuclear
*AaIAA16*	Aarg05G020910.1	265	29,066.46	8.78	37.29	80.11	−0.34	Nuclear
*AaIAA17*	Aarg06G011190.1	183	20,467.39	7.02	37.98	88.36	−0.431	Nuclear
*AaIAA18*	Aarg06G014790.1	190	21,153	8.3	45.9	57.47	−0.745	Nuclear
*AaIAA19*	Aarg06G021200.1	229	25,531.29	8.16	40.71	70.61	−0.512	Nuclear
*AaIAA20*	Aarg07G015030.1	155	18,391.91	5.46	58.75	80.45	−0.623	Nuclear
*AaIAA21*	Aarg07G020250.1	332	35,671.32	8.66	45.05	70.42	−0.48	Nuclear
*AaIAA22*	Aarg08G015630.1	207	22,925.83	5.97	32.7	81.4	−0.529	Nuclear
*AaIAA23*	Aarg08G015670.1	209	23,189.17	5.61	32.82	82.49	−0.489	Nuclear
*AaIAA24*	Aarg08G018190.1	313	34,615.93	6.4	57.62	68.79	−0.486	Nuclear
*AaIAA25*	Aarg09G003120.1	190	21,183.09	8.3	46.6	57.47	−0.731	Nuclear
*AaIAA26*	Aarg09G037610.1	654	72,827.47	6.04	55.49	71.68	−0.471	Nuclear
*AaIAA27*	Aarg10G000170.1	257	28,609.35	8.37	48.75	60.7	−0.654	Nuclear
*AaIAA28*	Aarg10G001790.1	254	28,301.06	8.87	48.75	60.67	−0.693	Nuclear
*AaIAA29*	Aarg10G002700.1	1044	117,163.7	6.56	70.7	68	−0.716	Nuclear
*AaIAA30*	Aarg10G004460.1	206	23,406.49	6.88	49.76	68.54	−0.712	Nuclear
*AaIAA31*	Aarg10G012770.1	179	20,275.95	5.97	62.44	63.69	−0.765	Nuclear
*AaIAA32*	Aarg10G013430.1	191	21,642.52	5.39	61.81	62.72	−0.702	Nuclear
*AaIAA33*	Aarg10G054010.1	238	26,322.34	8.59	29.37	72.02	−0.575	Nuclear
*AaIAA34*	Aarg10G056370.1	162	18,394.02	7.95	61.36	88.46	−0.577	Nuclear
*AaIAA35*	Aarg11G002340.1	241	26,451.1	9.11	41.04	75.98	−0.611	Nuclear
*AaIAA36*	Aarg11G002950.1	263	28,731.7	8.13	40.33	76.65	−0.393	Nuclear
*AaIAA37*	Aarg11G003340.1	169	18,728.47	6.31	41.01	77.81	−0.439	Nuclear
*AaIAA38*	Aarg11G003350.1	212	23,901.29	7.53	48.43	78.07	−0.412	Nuclear
*AaIAA39*	Aarg11G014830.1	326	34,716.01	8.14	43	71.81	−0.52	Nuclear
*AaIAA40*	Aarg11G030150.1	235	26,733.46	6.24	44.76	76.26	−0.397	Nuclear
*AaIAA41*	Aarg11G030160.1	197	22,766.72	6.39	50.71	63.71	−0.724	Nuclear
*AaIAA42*	Aarg12G024700.1	256	27,610.54	6.45	49.77	62.03	−0.674	Nuclear
*AaIAA43*	Aarg12G025240.1	255	27,696.68	6.45	49.29	63.8	−0.677	Nuclear
*AaIAA44*	Aarg13G015770.1	177	20,088.97	7.66	56.27	63.84	−0.69	Nuclear
*AaIAA45*	Aarg13G016280.1	217	23,777.13	5.87	43.99	87.51	−0.248	Nuclear
*AaIAA46*	Aarg13G016670.1	188	20,803.73	5.59	48.35	87.5	−0.393	Nuclear
*AaIAA47*	Aarg13G025060.1	229	25,974.51	4.96	40.33	73.58	−0.758	Nuclear
*AaIAA48*	Aarg14G002120.1	221	24,696.54	6.92	29.79	59.05	−0.854	Nuclear
*AaIAA49*	Aarg14G003710.1	222	24,697.49	6.92	28.42	57.93	−0.863	Nuclear
*AaIAA50*	Aarg14G012600.1	190	21,158.13	6.6	39.98	85.63	−0.453	Nuclear
*AaIAA51*	Aarg14G023130.1	229	25,503.23	8.16	40.34	69.78	−0.522	Nuclear
*AaIAA52*	Aarg15G014380.1	155	18,391.91	5.46	58.75	80.45	−0.623	Nuclear
*AaIAA53*	Aarg15G018930.1	332	35,698.35	8.66	43.89	70.42	−0.486	Nuclear
*AaIAA54*	Aarg15G024900.1	193	22,277.12	5.22	37.4	88.81	−0.439	Nuclear
*AaIAA55*	Aarg16G015470.1	214	23,544.67	5.95	32.54	86.45	−0.323	Nuclear
*AaIAA56*	Aarg16G015550.1	214	23,557.75	5.95	32.54	88.74	−0.288	Nuclear
*AaIAA57*	Aarg17G001730.1	190	21,183.09	8.3	46.6	57.47	−0.731	Nuclear
*AaIAA58*	Aarg0G001160.1	237	26,068.55	8.6	42.49	64.18	−0.673	Nuclear
*AaIAA59*	Aarg0G013210.1	175	19,772.52	5.2	61.38	71.83	−0.57	Nuclear
*AaIAA60*	Aarg0G017000.1	326	34,716.01	8.14	43	71.81	−0.52	Nuclear
*AaIAA61*	Aarg0G047700.1	155	18,391.91	5.46	58.75	80.45	−0.623	Nuclear

## Data Availability

Data are contained within the article and Appendix A.

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
