# Peer review of "Genome-Wide Analysis of Aux/IAA Gene Family in Artemisia argyi: Identification, Phylogenetic Analysis, and Determination of Response to Various Phytohormones"

_plants, 2024, doi:10.3390/plants13050564_

Round 1

Reviewer 1 Report

Comments and Suggestions for Authors

Reviewer comments:
The manuscript ID: plants-2805584 entitled “Genome-Wide Analysis of Aux/IAA Gene Family in Artemisia argyi: Identification, Phylogenetic Analysis and Determination of Response to Different Phytohormones” by Lian et al. I found this topic interesting, demonstrates, genome-wide analysis of Aux/IAA gene family in A. argyi. But I have few concerns related to the research article. I am asking authors to revise the manuscript carefully considering my comments for possible publication in “plants”.
I have given my comments.

Abstract:
• Line No 20: Authors must check and correct the sentence: “Artemisiae argyi is a traditional herbal medicine” medicinal plant.
• The main feature of the research study and how it can impact future research should be highlighted in the abstract.

Introduction:
• Introduction is well followed. Authors must give some of the area of cultivation, production information and yield information of Artemisia.

Materials and methods:
• What is the specific reason for using, “Aseptic tissue culture seedlings for analysis”.
• Line No 598: The authors requested to check and replace ‘1 μg qualified RNA’ with ‘quantified’.

Result and Discussion:
• Line No 117: Authors mentiomed, “AaIAAs location in chromosomes (Table 2)”. I am unable to find Table 2.
• Figure 8: Authors showing 3 replicates of leaf and root. Why is expression pattern not matching in replicates?
• Authors must check and remove the line spacing 643, 644.

• Authors have presented their finding in well manner with appropriate discussions, but I still not satisfied with recent works carried out on similar area. Most of the cited articles in the discussion section are not sufficient. It is suggested to improve the manuscript considering this specific aspect. I wish to see a revised version of the manuscript with a good discussion throughout the result and discussion section with recent. I have not found any reason how Aux/IAA will affect the yield of medicinal properties and secondary metabolites is not discussed in sufficient way.

In current format, I am not recommending the article for the publication but if authors can take serious efforts to improve the manuscript, I will be happy to re-review the manuscript.

Author Response

Dear Reviewer,

We appreciate you very much for your earnest and responsible to work, and provided positive and constructive comments and suggestions on our manuscript ID: plants-2805584 “Genome-Wide Analysis of Aux/IAA Gene Family in Artemisia argyi: Identification, Phylogenetic Analysis and Determination of Response to Different Phytohormones”. Those comments are all valuable and very helpful for revising and improving our manuscript, as well as the important guiding significance to our researches. We have considered each of the comments carefully and the itemized responses are outlined below. For the editing of English, we employed an English-language editing service. The paper has been professionally edited for English usage, grammar, spelling, and punctuation by a native English speaker and a skilled professional editor. Certification is attached at the end of this letter.

Moreover, we uploaded the file of a marked-up copy of our manuscript labeled 'Revised Manuscript with Track Changes', and a response letter that responds to each point raised by the academic editor and reviewer(s) labeled 'Response to Reviewers'. We believe that the new manuscript has been significantly improved by the suggested revised. We hope this revision can make our paper more acceptable in plants. The revisions were addressed point by point below:

Reviewer 1 comments:

The manuscript ID: plants-2805584 entitled “Genome-Wide Analysis of Aux/IAA Gene Family in Artemisia argyi: Identification, Phylogenetic Analysis and Determination of Response to Different Phytohormones” by Lian et al. I found this topic interesting, demonstrates, genome-wide analysis of Aux/IAA gene family in A. argyi. But I have few concerns related to the research article. I am asking authors to revise the manuscript carefully considering my comments for possible publication in “plants”.

I have given my comments.

Abstract:

  • Line No 20: Authors must check and correct the sentence: “Artemisiae argyi is a traditional herbal medicine” medicinal plant.

R: Thank you very much for you carefully advice. This sentence has been checked and corrected, and our paper has been professionally edited for English usage, grammar, spelling, and punctuation by a native English speaker and a skilled professional editor.

  • The main feature of the research study and how it can impact future research should be highlighted in the abstract.

R: Thank you very much for you valuable advice. The main feature of the research study and how it can impact future research had been highlighted in the abstract.

Introduction:

  • Introduction is well followed. Authors must give some of the area of cultivation, production information and yield information of Artemisia.

R: Thank you very much for you valuable advice. The information of the area of cultivation, production information and yield information of Artemisia has been given in the Introduction section in line 47-49, 57-59.

Materials and methods:

  • What is the specific reason for using, “Aseptic tissue culture seedlings for analysis”.

R: Thank you very much for you valuable question. For the Aseptic tissue culture seedlings, the growth of tissue culture seedlings is more consistent, and the growth environment is controllable, and it is not affected by the external environment, which is suitable for single factor treatment. In addition, the treatment of tissue culture seedlings in culture bottles can reduce the volatilization of hormone solutions, which is conducive to the absorption of hormones by plants

  • Line No 598: The authors requested to check and replace ‘1 μg qualified RNA’ with ‘quantified’.

R: Thank you very much for you carefully advice. This sentence has been checked and corrected, and this paper has been professionally edited for English usage, grammar, spelling, and punctuation by a native English speaker and a skilled professional editor.

Result and Discussion:

  • Line No 117: Authors mentiomed, “AaIAAs location in chromosomes (Table 2)”. I am unable to find Table 2.

R: Thank you very much for you carefully advice. (Table 2) has been revised to (Figure 1).

  • Figure 8: Authors showing 3 replicates of leaf and root. Why is expression pattern not matching in replicates?

R: Thank you very much for you valuable question. As can be seen from the Figure 7A (Figure 8) in revised manuscript, the expression level of most AaIAA genes is matched in replicates. While a few AaIAA genes are not matched in replicates, such as AaIAA7, AaIAA25, because these genes were extremely low abundance with FPKM<1, and these genes FPKM values were poor stability. In Figure 7A, the FPKM values in the heatmap were normalized by Z-score, its helpful to compare the expression levels in roots and leaves, but cannot show the expression levels between different AaIAA genes. So, we provide the raw FPKM values in Table S5.

  • Authors must check and remove the line spacing 643, 644.

R: Thank you very much for you carefully advice. The line spacing 643, 644, has been checked and removed.

  • Authors have presented their finding in well manner with appropriate discussions, but I still not satisfied with recent works carried out on similar area. Most of the cited articles in the discussion section are not sufficient. It is suggested to improve the manuscript considering this specific aspect. I wish to see a revised version of the manuscript with a good discussion throughout the result and discussion section with recent. I have not found any reason how Aux/IAA will affect the yield of medicinal properties and secondary metabolites is not discussed in sufficient way.

R: Thank you very much for you valuable advice. The discussion of “Aux/IAA will affect the yield of medicinal properties and secondary metabolites” was added in the 3.4 section in the revised manuscript. As follows:

3.4. Possible correlation between Aux/IAA gene and switching to secondary metabolism in A. argyi

Previous studies have shown that various stresses and hormones can regulate the accumulation of secondary metabolites, especially in medicinal plants [54,55]. In Glycyrrhiza uralensis, exogenous ABA can promote the the synthesis of active ingredients, including isoliquiritin, glycyrrhizin, liquiritigenin, liquiritin and isoliquiritigenin [54]. In Ruta graveolens, salicylic acid can promoter the accumulation of phenols and flavonoids in callus [55]. In cultured Onosma paniculatum cells, brassinolide together with IAA (indoleacetic acid) and BAP (6-benzylaminopurine) can enhance the accumulation of shikonin at appropriate concentrations [56]. In apple, it has been reported that auxin regulates anthocyanin biosynthesis through the Aux/IAA-ARF signaling pathway [57]. In A. argyi, previous studies have shown that exogenous hormone (MeJA, SA and ABA) treatments changed the eight secondary metabolites, including phenylpropanoids, flavonoids, terpenoids, alkaloids and others [58]. In our study, the expression levels of 12 AaIAA genes changed in various degrees in responses to different hormones (IAA, SA, MeJA, and ABA), combining with the regulation of different hormones on the accumulation of secondary metabolites in A. argyi, it is speculated that MeJA, SA, IAA, and ABA regulated the accumulation of the secondary metabolites in A. argyi may throught the Aux/IAA genes or Aux/IAA-related signaling pathway. In addition, Aux/IAA proteins, act as an important component of auxin signaling pathway, play a key regulatory role in the process of plant growth [5]. The accumulation of plant secondary metabolites is always in a dynamic process, which is also constantly changing in the process of plant growth [59]. It is suggested that AaIAA proteins may play a role in mediating plant growth and the accumulation of secondary metabolites. Above all, the regulation of secondary metabolites accumulation by Aux/IAA protein in A. argyi needs to be further studied.

Once again, thank you very much for your work and valuable comments.  

Kind regards, â€¯

Conglong Lian, Jinxu Lan, Rui Ma, Jingjing Li, Fei Zhang, Bao Zhang, Xiuyu Liu, Suiqing Chen*

Reviewer 2 Report

Comments and Suggestions for Authors

The manuscript is well organized and fluently written. Overall, I like the manuscript and approve for publication after addressing following concerns/suggestion:

·       Line 114, its better to full name first and later abbreviation., also include reference for all the software and methodology used during analysis, such as HHMER

·       Line 116, they identified in the A. argyi ? genome or what? Any reference or link

·       Line 135, it s better not to start a sentence form a number, so rephrased it

·       Line 167-168, the botanical name should be italicized

·       Line 180-185, authors are talking about gene structure (CDS) so, gene name should be italicized

·       Figure 5, I suggest add phylo tree and link together

·       Combine figure 8 and 9

·       Combine figure 3 and five together, there is no need to be a greater number of figures but compact and sound study is better

·       Line 570, A total of not total 21

Author Response

Dear Reviewer,

We appreciate you very much for your earnest and responsible to work, and provided positive and constructive comments and suggestions on our manuscript ID: plants-2805584 “Genome-Wide Analysis of Aux/IAA Gene Family in Artemisia argyi: Identification, Phylogenetic Analysis and Determination of Response to Different Phytohormones”. Those comments are all valuable and very helpful for revising and improving our manuscript, as well as the important guiding significance to our researches. We have considered each of the comments carefully and the itemized responses are outlined below. For the editing of English, we employed an English-language editing service. The paper has been professionally edited for English usage, grammar, spelling, and punctuation by a native English speaker and a skilled professional editor. Certification is attached at the end of this letter.

Moreover, we uploaded the file of a marked-up copy of our manuscript labeled 'Revised Manuscript with Track Changes', and a response letter that responds to each point raised by the academic editor and reviewer(s) labeled 'Response to Reviewers'. We believe that the new manuscript has been significantly improved by the suggested revised. We hope this revision can make our paper more acceptable in plants. The revisions were addressed point by point below:

Comments and Suggestions for Authors:

The manuscript is well organized and fluently written. Overall, I like the manuscript and approve for publication after addressing following concerns/suggestion:

  • Line 114, its better to full name first and later abbreviation., also include reference for all the software and methodology used during analysis, such as HHMER

R: Thank you very much for you valuable advice. This kind of problem has been revised in the paper, including HHMER, BLASTP, MEME, FPKM, RT-qPCR, MeJA, SA, ABA, IAA, etc.

  • Line 116, they identified in the A. argyi ? genome or what? Any reference or link

R: Thank you very much for you valuable advice. This sentence has been revised to “a total of 61 Aux/IAA members were identified in the genome data of A. argyi (Table 1) [30]”, and the reference [30] was added.

  • Line 135, it s better not to start a sentence form a number, so rephrased it

R: Thank you very much for you valuable advice. This kind of issue has been revised.

  • Line 167-168, the botanical name should be italicized

R: Thank you very much for you carefully advice. This question has been revised in line 171-176.

  • Line 180-185, authors are talking about gene structure (CDS) so, gene name should be italicized

R: Thank you very much for you carefully advice. This question has been revised in line 187-194.

  • Figure 5, I suggest add phylo tree and link together
  • Combine figure 8 and 9
  • Combine figure 3 and five together, there is no need to be a greater number of figures but compact and sound study is better
  1. R.Thank you very muchfor you valuable advice. Figure 3 and 5 were combine into “Figure 3” in the revised manuscript; Figure 8 and 9 were combine into “Figure 7” in the revised manuscript.
  • Line 570, A total of not total 21
  1. Thank you very muchfor you carefully advice. “A total 21” was revised to “A total of 21” in the revised manuscript in line 621.

Once again, thank you very much for your work and valuable comments.  

Kind regards, â€¯

Conglong Lian, Jinxu Lan, Rui Ma, Jingjing Li, Fei Zhang, Bao Zhang, Xiuyu Liu, Suiqing Chen*

Reviewer 3 Report

Comments and Suggestions for Authors

The current paper describe structure and function of the Aux/IAA gene in medical plants Artemesia.

 Despite the results may some contribution to the field, many clarifications are required.

Below are some points.

Please, come through text and make sentences more clear, avoid multi-messages one..

Line 20: : “Artemisiae argyi is a traditional herbal medicine” ¿? Plants is not medicine itself.

Line 36: “some AaIAAs involved in salt and saline-alkali stresses” ¿?? Gene can not be invloved itself in stress. They can changes expression level under stress. Morever, see below: you did not desribe what do you mean as su kinds of stresses.  

Lines 45 – 47: it will be nice to have at least one link to Artemisia plant as citation.

Line 52: “A. argyi can be applied to treatment” ¿?? Plants itself can not be applied. Extract can be applied…

Lines 57 -61_ long sentence with confusing messages…

Line 61: somehow confusing auxin moelcule with protein. Aux/IAA protein should be.

Line 81: regulation of diverse organs” ¿??

Lines 316 – 338: authors decribed results of gene expression under salt and saline-alkaline stress. However, autors did not describe how do they induced such stress what did they stress ean. This part can not be evelauted in the absence of such information.

Fig 10: why in the case of SA legende panle was on left side, while in others in the right side?

Panel A can not be compare with panel B, C and D because the way of treatment is different. Moreover, aux/IAA gene exopresed differently in different cell type and show different kinetics as well.even in palisade mesophyl and in vasculature, for example. This should be mentioned.

Lines 529-530: please, edit sentence. Olease, provide reference that A-argyi realy require N:P ratio 40:1 and halogen as main cation (not P).

If plant growth from seeds, please, provide seeds source and sterlization protocol.  Moreover, I am sure that whole roo mis sterile. Sterile is only bottle inside.

Please, provide evidences that you used specifci concentration of SA, MeJ, ABA and IAA. You know that all treatments with high concentration caused non-specific effects  and can not be consider as only specifci hormones. The doce -response curve must be done as first step.

Moreover, leaf have a cuticle nwhcih prevent watr uptake. So, how effcetive were such kind of treatemnts ea. what was the real effective concentration of vSA in mesophyl cell?

It will be nice alos discuss the possible correlation between Aux/IAA gene and switching to secondary metabolism. Oil accumulation, for example, is well-know  process of the secondary metabolism. Auixn, SA. MeJ (even in non-specific “window” definitely can switch to secondary metabolism in developed organs like leaves.

Authors need to mention and discuss this.  

Comments on the Quality of English Language

Please, check carefully, avoid long sentence with multi-messages in one sentence.

Author Response

Dear Reviewer,

We appreciate you very much for your earnest and responsible to work, and provided positive and constructive comments and suggestions on our manuscript ID: plants-2805584 “Genome-Wide Analysis of Aux/IAA Gene Family in Artemisia argyi: Identification, Phylogenetic Analysis and Determination of Response to Different Phytohormones”. Those comments are all valuable and very helpful for revising and improving our manuscript, as well as the important guiding significance to our researches. We have considered each of the comments carefully and the itemized responses are outlined below. For the editing of English, we employed an English-language editing service. The paper has been professionally edited for English usage, grammar, spelling, and punctuation by a native English speaker and a skilled professional editor. Certification is attached at the end of this letter.

Moreover, we uploaded the file of a marked-up copy of our manuscript labeled 'Revised Manuscript with Track Changes', and a response letter that responds to each point raised by the academic editor and reviewer(s) labeled 'Response to Reviewers'. We believe that the new manuscript has been significantly improved by the suggested revised. We hope this revision can make our paper more acceptable in plants. The revisions were addressed point by point below:

Comments and Suggestions for Authors:

The current paper describe structure and function of the Aux/IAA gene in medical plants Artemesia.

Despite the results may some contribution to the field, many clarifications are required.

Below are some points.

Please, come through text and make sentences more clear, avoid multi-messages one.

R: Thank you very much for you valuable advice. We had come through the article several times, and our paper has been professionally edited for English usage, grammar, spelling, and punctuation by a native English speaker and a skilled professional editor.

Line 20: : “Artemisiae argyi is a traditional herbal medicine” ¿? Plants is not medicine itself.

R: Thank you very much for you valuable advice. This sentence has been revised to “Artemisia argyi is a traditional herbal medicine plant, and its folium artemisia argyi is widely in demand due to moxibustion applications globally.”

Line 36: “some AaIAAs involved in salt and saline-alkali stresses” ¿?? Gene can not be invloved itself in stress. They can changes expression level under stress. Morever, see below: you did not desribe what do you mean as su kinds of stresses.  

R: Thank you very much for you valuable advice. This sentence was revised to “some AaIAAs were involved in the regulation of salt and saline-alkali stresses.” The description of su kinds of stresses were added in the 4.1 section in line 582-591.

Lines 45 – 47: it will be nice to have at least one link to Artemisia plant as citation.

R: Thank you very much for you valuable advice. One link to Artemisia plant (https://www.plantplus.cn/info/Artemisia%20argyi?t=foc) was added as citation in the revised manuscript.

Line 52: “A. argyi can be applied to treatment” ¿?? Plants itself can not be applied. Extract can be applied…

R: Thank you very much for you valuable advice. “A. argyi” has been revised to “folium artemisiae argyi”. “folium artemisiae argyi” is a traditional Chinese medicine name of “A. argyi” .

Lines 57 -61_ long sentence with confusing messages…

R: Thank you very much for you valuable advice. This long sentence has been revised to “Plant hormones are critical for the regulation of plant growth and development in various stages, especially auxin. Many of these rely on the auxin signaling pathway, which determines the fate of plants from birth to death and plays essential roles during plant life cycles, including embryogenesis, cell division and elongation, stress response, and secondary metabolite biosynthesis ”. And our paper has been professionally edited for English usage, grammar, spelling, and punctuation by a native English speaker and a skilled professional editor.

Line 61: somehow confusing auxin moelcule with protein. Aux/IAA protein should be.

R: Yes, It's Aux/IAA protein, thank you very much for you valuable advice. And this sentence has been revised to “Auxin/indole-3-acetic acid (Aux/IAA, or IAA) proteins, are important components of the auxin signaling pathway” in line 65.

Line 81: regulation of diverse organs” ¿??

R: Thank you very much for you valuable advice. This sentence has been revised to “in the regulation of the development of diverse organs ”.

Lines 316 – 338: authors decribed results of gene expression under salt and saline-alkaline stress. However, autors did not describe how do they induced such stress what did they stress ean. This part can not be evelauted in the absence of such information.

R: Thank you very much for you valuable advice. The description of salt and saline-alkaline stresses were added in the 4.1 section in line 582-591.

Fig 10: why in the case of SA legende panle was on left side, while in others in the right side?

R: Thank you very much for you carefully advice. Neglect in the drawing to unity in the case of SA legende panle, and this issue has been modified in the revised manuscript in Figure 8.

Panel A can not be compare with panel B, C and D because the way of treatment is different. Moreover, aux/IAA gene exopresed differently in different cell type and show different kinetics as well.even in palisade mesophyl and in vasculature, for example. This should be mentioned.

R: Thank you very much for you valuable advice. Yes, panel A, B, C, and D are compared independently. We only studied its expression patterns in leaves, not carry out Aux/IAA gene expressed in different cell type. This is a question worthy of further study, and points out the direction for our follow-up research.

Lines 529-530: please, edit sentence. Olease, provide reference that A-argyi realy require N:P ratio 40:1 and halogen as main cation (not P).

R: Thank you very much for you carefully advice. The culture medium conditions of A. argyi were tested in the early stage, and the references [60] was added in the revised manuscript in line 571.

If plant growth from seeds, please, provide seeds source and sterlization protocol.  Moreover, I am sure that whole room is sterile. Sterile is only bottle inside.

R: Thank you very much for you valuable advice. These information were added in 4.1 section in line 565-568. “The plant materials in our study were obtained from the Medicinal Botanical Garden of Henan University of Chinese Medicine. The root buds of A. argyi were used as explants for disinfected, and sterile tissue culture seedlings of A. argyi were obtained as described by Li, et al. [60].”

Please, provide evidences that you used specifci concentration of SA, MeJ, ABA and IAA. You know that all treatments with high concentration caused non-specific effects  and can not be consider as only specifci hormones. The doce -response curve must be done as first step.

R: Thank you very much for you valuable advice. After summarizing many references, the concentration of hormones (ABA, SA, and MeJA) treatment used generally rang from10 to 500 µM. Referring to the ABA concentration applied in this article [Ke, Y.; Abbas, F.; Zhou, Y.; Yu, R.; Yue, Y.; Li, X.; Yu, Y.; Fan, Y. Genome-Wide Analysis and Characterization of the Aux/IAA Family Genes Related to Floral Scent Formation in Hedychium coronarium. Int. J. Mol. Sci. 2019, 20, 3235.], the hormones concentration of 200 µM was also selected for experimental treatment in our study. For the applied concentration of IAA, most articles use 10 µm concentration, such in Brassica rapa [Paul P, Dhandapani V, Rameneni J J, et al. Genome-wide analysis and characterization of Aux/IAA family genes in Brassica rapa. PloS one, 2016, 11(4): e0151522.], and Tartary Buckwheat[Yang F, Zhang X, Tian R, et al. Genome-Wide Analysis of the Auxin/Indoleacetic Acid Gene Family and Response to Indole-3-Acetic Acid Stress in Tartary Buckwheat (Fagopyrum tataricum). International Journal of Genomics, 2021, 2021,3102399.]. This is a shortcoming of our paper, for the selection of hormone concentrations, the dose-response curve was not done as first step. But we believe that this study explored the response patterns of different AaIAA members at this concentration in response to different hormones, and to some extent reflected the differential expression of different AaIAA members in response to different hormones.

Moreover, leaf have a cuticle nwhcih prevent watr uptake. So, how effcetive were such kind of treatemnts ea. what was the real effective concentration of vSA in mesophyl cell?

R: Thank you very much for you valuable advice. This is a problem worthy of deep thinking and research. In our study, tissue culture seedlings were selected for hormone treatment, and the tissue culture seedlings were relatively young with less cuticle content. In addition, we treated them in a closed tissue culture bottle, which reduced the volatilization of hormone solution in the air and was more conducive to the absorption of plants. Therefore, the experiment of this study is relatively rigorous, and its real effective concentration of different hormones in mesophyl cell remains to be detected, which provides valuable suggestions for our next study.

It will be nice also discuss the possible correlation between Aux/IAA gene and switching to secondary metabolism. Oil accumulation, for example, is well-know  process of the secondary metabolism. Auixn, SA. MeJ (even in non-specific “window” definitely can switch to secondary metabolism in developed organs like leaves. 

Authors need to mention and discuss this.  

R: Thank you very much for you valuable advice. The discuss of “correlation between Aux/IAA gene and switching to secondary metabolism” was added in the 3.4 section in the revised manuscript. As follows:

3.4. Possible correlation between Aux/IAA gene and switching to secondary metabolism in A. argyi

Previous studies have shown that various stresses and hormones can regulate the accumulation of secondary metabolites, which can regulate the quality of medicinal plants [54,55]. In Glycyrrhiza uralensis, exogenous ABA can promote the the synthesis of active ingredients, including isoliquiritin, glycyrrhizin, liquiritigenin, liquiritin and isoliquiritigenin [54]. In Ruta graveolens, salicylic acid can promoter the accumulation of phenols and flavonoids in callus [55]. In cultured Onosma paniculatum cells, brassinolide together with IAA (indoleacetic acid) and BAP (6-benzylaminopurine) can enhance the accumulation of shikonin at appropriate concentrations [56]. In apple, it has been reported that auxin regulates anthocyanin biosynthesis through the Aux/IAA-ARF signaling pathway [57]. In A. argyi, previous studies have shown that exogenous hormone (MeJA, SA and ABA) treatments changed the eight secondary metabolites, including phenylpropanoids, flavonoids, terpenoids, alkaloids and others [58]. In our study, the expression levels of 12 AaIAA genes changed in various degrees in responses to different hormones (IAA, SA, MeJA, and ABA), combining with the regulation of different hormones on the accumulation of secondary metabolites in A. argyi, it is speculated that MeJA, SA, IAA, and ABA regulated the accumulation of the secondary metabolites in A. argyi may throught the Aux/IAA genes or Aux/IAA-related signaling pathway. In addition, Aux/IAA proteins, act as an important component of auxin signaling pathway, play a key regulatory role in the process of plant growth [5]. The accumulation of plant secondary metabolites is always in a dynamic process, which is also constantly changing in the process of plant growth [59]. It is suggested that AaIAA proteins may play a role in mediating plant growth and the accumulation of secondary metabolites. Above all, the regulation of secondary metabolites accumulation by Aux/IAA protein in A. argyi needs to be further studied.

Comments on the Quality of English Language

Please, check carefully, avoid long sentence with multi-messages in one sentence.

  1. R.Thank you very muchfor you valuable advice. We had check the full paper carefully, and our paper has been professionally edited for English usage, grammar, spelling, and punctuation by a native English speaker and a skilled professional editor.

Once again, thank you very much for your work and valuable comments. 

Kind regards, â€¯

Conglong Lian, Jinxu Lan, Rui Ma, Jingjing Li, Fei Zhang, Bao Zhang, Xiuyu Liu, Suiqing Chen*

Round 2

Reviewer 1 Report

Comments and Suggestions for Authors

Reviewer comments:

I have carefully read the manuscript ID: (plants-2805584) entitled “Genome-Wide Analysis of Aux/IAA Gene Family in Artemisia argyi: Identification, Phylogenetic Analysis and Determination of Response to Different Phytohormones” by Lian et al. This work presents interesting results on Genome wide analysis of Aux/IAA gene family in Artemisia argyi., and the authors have presented their findings in well manner with appropriate figures.  

Authors addressed all the comments and submitted the revised version. Overall, it is an interesting study and should be considered for publication in Plant.

Author Response

Response to Reviewers

Dear Reviewer,

We appreciate you very much for your earnest and responsible to work, and provided positive and constructive comments and suggestions on our manuscript ID: plants-2805584 “Genome-Wide Analysis of Aux/IAA Gene Family in Artemisia argyi: Identification, Phylogenetic Analysis and Determination of Response to Various Phytohormones”. Those comments are all valuable and very helpful for revising and improving our manuscript, as well as the important guiding significance to our researches.

Reviewer comments:

I have carefully read the manuscript ID: (plants-2805584) entitled “Genome-Wide Analysis of Aux/IAA Gene Family in Artemisia argyi: Identification, Phylogenetic Analysis and Determination of Response to Different Phytohormones” by Lian et al. This work presents interesting results on Genome wide analysis of Aux/IAA gene family in Artemisia argyi., and the authors have presented their findings in well manner with appropriate figures.  

Authors addressed all the comments and submitted the revised version. Overall, it is an interesting study and should be considered for publication in Plant.

R: Thank you very much. We greatly appreciate your hard work and responsibility in providing positive and constructive feedback and recommendations on our manuscript. And we all believe that the proposed revisions have substantially improved our new manuscript. Once again, thank you very much.

Once again, thank you very much for your work and valuable comments.  

Kind regards, â€¯

Conglong Lian, Jinxu Lan, Rui Ma, Jingjing Li, Fei Zhang, Bao Zhang, Xiuyu Liu, Suiqing Chen*

Reviewer 3 Report

Comments and Suggestions for Authors

Thank you, you made a great work for improve the text.

There are some more corrections require.

Line 435_ different, not differentiated.

Line 571: sterile culture room?? Plant growth under sterile conditions, not in sterile room.

Line 587: 200 mM each or total?

Authors still not explain how halogen treatments (according to citation 60) interact with hormone effect. https://www.mdpi.com/2223-7747/13/2/327

Actually, it is not enough only cited one-two paper insted make dose-response curve. Different species, plant age, conditions definitely changes such a curve.

Comments on the Quality of English Language

Several polishing still require.

Round 3

Reviewer 3 Report

Comments and Suggestions for Authors

Thank you!

Almost all corrections was done. However, one importnat point still missing: authors need to be very carefully with lines 577 -578- MS medium is far form opptimal for nutrients ratio and contains quite toxic halogen in high concentration. Please, next time use normal medium

Author Response

Dear Reviewer,

We appreciate you very much for your earnest and responsible to work, and provided positive and constructive comments and suggestions on our manuscript ID: plants-2805584 “Genome-Wide Analysis of Aux/IAA Gene Family in Artemisia argyi: Identification, Phylogenetic Analysis and Determination of Response to Various Phytohormones”. Those comments are all valuable and very helpful for revising and improving our manuscript, as well as the important guiding significance to our researches. We have considered each of the comments carefully and the itemized responses are outlined below.

Comments and Suggestions for Authors

Almost all corrections was done. However, one importnat point still missing: authors need to be very carefully with lines 577 -578- MS medium is far form opptimal for nutrients ratio and contains quite toxic halogen in high concentration. Please, next time use normal medium

R: Thank you very much for you valuable advice. It provides great suggestion for our next time research. This sentence has been corrected at line 577, or see below.

“The medium has a high concentration of inorganic salts and ions, also including Iodine and Chloride, is a relatively stable ion balance solution, has a high nitrate content, the right amount and proportion of nutrients, and can meet the nutritional and physiological needs of plant cells [61].” was revised to “The medium has a high concentration of inorganic salts and ions, also including Iodine and Chloride, is a relatively stable ion balance solution, has a high nitrate content, and can meet the nutritional and physiological needs of most plant cells [61].”

Once again, thank you very much for your work and valuable comments.  

Kind regards, â€¯

Conglong Lian, Jinxu Lan, Rui Ma, Jingjing Li, Fei Zhang, Bao Zhang, Xiuyu Liu, Suiqing Chen*